# Designing Gallium-Containing Hydroxyapatite Coatings on Low Modulus Beta Ti-45Nb Alloy



Jithin Vishnu [1,2,3] , Andrea Voss [1], Volker Hoffmann [1], Ludovico Andrea Alberta [1] , Adnan Akman [1] ,
Balakrishnan Shankar [2,3], Annett Gebert [1] and Mariana Calin [1,*]

1   Leibniz Institute for Solid State and Materials Research Dresden (IFW Dresden), Helmholtzstr. 20,
    D-01069 Dresden, Germany; jithinv@am.amrita.edu (J.V.); a.voss@ifw-dresden.de (A.V.);
    v.hoffmann@ifw-dresden.de (V.H.); l.a.alberta@ifw-dresden.de (L.A.A.); a.akman@ifw-dresden.de (A.A.);
    a.gebert@ifw-dresden.de (A.G.)
2   Centre for Flexible Electronics and Advanced Materials, Amrita Vishwa Vidyapeetham,
    Amritapuri 690525, Kerala, India; bala@am.amrita.edu
3   Department of Mechanical Engineering, Amrita Vishwa Vidyapeetham, Amritapuri 690525, Kerala, India
*   Correspondence: m.calin@ifw-dresden.de; Tel.: +49-351-4659-613

**Abstract:** Low-modulus β-type Ti-45Nb alloy is a promising implant material due to its good mechanical biocompatibility, non-toxicity, and outstanding corrosion resistance. Its excellent chemical stability brings new challenges to chemical surface modification treatments, which are indispensable for both osteogenesis and antibacterial performance. Coatings containing metal ions as anti-microbial agents can be an effective way to reduce implant-associated infections caused by bacterial biofilm. Gallium ion ($Ga^{3+}$) has the potential to reduce bacterial viability and biofilm formation on implant surfaces. In this study, a novel two-step process has been proposed for $Ga^{3+}$ incorporation in hydroxyapatite (HAP) to develop bioactive and antibacterial surfaces on Ti-45Nb alloy. For the generation of bioactive surface states, HAP electrodeposition was conducted, followed by wet chemical immersion treatments in gallium nitrate (1 mM). Different buffers such as phosphate, sodium bicarbonate, ammonium acetate, and citrate were added to the solution to maintain a pH value in the range of 6.5–6.9. Coating morphology and HAP phases were retained after treatment with gallium nitrate, and $Ga^{3+}$ ion presence on the surface up to 1 wt.% was confirmed. Combining Ga and HAP shows great promise to enable the local delivery of $Ga^{3+}$ ions and consequent antibacterial protection during bone regeneration, without using growth factors or antibiotics.

**Keywords:** hydroxyapatite; antibacterial gallium; electrodeposition; coating; low modulus beta titanium





## 1. Introduction

Titanium and its alloys are widely used for orthopedic load-bearing implant applications owing to their proven biocompatibility, excellent corrosion resistance, and good combination of mechanical properties as compared to conventional 316 L stainless steel and cobalt-chrome alloys [1,2]. Among the various Ti-based materials, beta Ti-Nb alloys are garnering immense research attention owing to their lower elastic modulus (E) values and biocompatible compositions [3]. However, similar to other widely used metallic biomaterials, the Ti alloy surface is bioinert, resulting in insufficient osseointegration. The Ti surface (with a native oxide layer) is incapable of chemically or biologically bonding with surrounding tissues in the peri-implant region, which eventually results in inferior cellular interactions after implantation [4,5]. Moreover, the biomaterial surface is prone to bacterial adhesion and subsequent biofilm formation, which can lead to the development of prosthetic infections [6]. These problems can in turn lead to the necessity of prolonged antibiotic therapy and eventually to the removal of the implant, with a consequent significant increase in hospitalization times and costs, together with a stressful, painful, and critical situation for the patient [7]. Improving the bonding performance of implant material with

the surrounding bone tissue by mitigating the inflammatory response has become a hot research topic, thus raising the success rate of implantation. The application of surface coatings is one of the most frequently used methods for the fabrication of bioactive surfaces with antibacterial properties. For instance, the incorporation of antibacterial agents (e.g., Ag, Cu, Zn, and Ga ions) into titanium bioactive-designed surfaces offers a potential solution.

Hydroxyapatite (HAP), with the chemical formula $Ca_{10}(PO_4)_6(OH)_2$, is the closest pure synthetic equivalent to human bone mineral. It is biocompatible, bioactive (capable of stimulating biological responses), and osteoconductive (permits bone ingrowth into its porous-structured surface) [8]. HAP is one of the most widely used bioceramic coating materials for orthopedic and dental implants and is used for bone regeneration therapies [9,10]. There are several well-developed technologies for depositing HAP on metallic substrates, including but not limited to thermal spray techniques [11,12], pulsed laser deposition [13], ion-beam-assisted deposition [14], electrochemical deposition [15,16], electrophoretic deposition [17], sol–gel technique [18], hydrothermal synthesis [19], etc. Among the various techniques, electrodeposition yields a number of benefits such as low cost, time effectiveness, better control of coating thickness, composition, and morphology, ability to coat non-line-of-sight surfaces, porous/geometrically complex tiny structures, and lower operating temperatures which can yield a highly crystalline deposit with low solubility in physiological fluids (higher operating temperatures are undesirable as it can alter substrate material microstructure-property aspects) [16,20,21].

For the purpose of imparting antibacterial activity, HAP is often incorporated with potential antibacterial elements such as Ag, Cu, Zn, and Ga [22]. Among these elements, Ga represents a prospective therapeutic incorporation into HAP as it is antibacterial, antiresorptive, antineoplastic against certain cancers, capable of hindering bone resorption, able to promote osteogenesis, and widely used as a bone tumor imaging agent [23,24]. The possibility of using this element to properly dope the surface of implantable devices by different surface modification techniques opens new perspectives for the prevention of infections associated with bone-related implants. The antibacterial potential of $Ga^{3+}$ lies in its ability to function as a ferric ion ($Fe^{3+}$) mimetic by disrupting microbial iron uptake [25]. Our group has recently developed novel Ti-45Nb-(2-8)Ga (wt.%) alloys and investigated the mechanical [26], tribocorrosion [27], electronic parameters (ab initio) [28], passivity, and antibacterial aspects [29].

Several gallium-based compounds have been explored for clinical applications, which include gallium nitrate, gallium maltolate, gallium thiosemicarbazone, tris(8-quinolinolato) gallium(III), and gallium citrate [25,30]. The incorporation of $Ga^{3+}$ ions into HAP coating has unfolded new research avenues for the theranostics of bone-tissue-related problems. To develop Ga-containing calcium phosphates, various chemical synthesis routes, such as sol–gel technique ($Ga^{3+}$-doped amorphous calcium phosphate nanoparticles [31]), wet chemical precipitation (HAP-Ga powder by co-precipitation, transformation, and ion-exchange [32,33]), dry method (heating mixture of reactants to form $Ga^{3+}$ enriched HAP powders [34], Ga-doped β-tricalcium phosphate [35,36]), have been reported. Melnikov et al. [37] reported the introduction of Ga ions into the HAP matrix by doping the prepared HAP with Ga in the form of gallium nitrate and sodium gallate.

One of the main limitations associated with a single-step chemical procedure to develop Ga-containing pure HAP is the difficulty of maintaining a pH value range, favorable for HAP development. $Ga^{3+}$ ions can undergo nearly complete hydrolyzation over a wide pH value range forming various hydroxides (particularly gallate $Ga(OH_4)^-$) and consequently, hydronium ion ($H_3O^+$) generation renders the solution more acidic [38,39]. This will potentially affect the development and stability/integrity of the HAP coating. The other problem is that, during the HAP electrodeposition process, hydroxide ions are cathodically generated, which are required to induce the (chemical) precipitation of HAP. $Ga^{3+}$ species, if any are present in the electrolyte for HAP electrodeposition, could transform into a soluble Ga-hydroxide and could be lost for the coating process. In order to overcome these limitations, in this study, various buffers were screened to keep the pH

stable (to maintain the Ga ions in the electrolyte) and ensure the integrity of the HAP coating implicitly.

The major aim of the present work is to explore the possibilities of incorporating $Ga^{3+}$ ions into electrodeposited HAP coatings on a low-modulus Ti-45Nb alloy to provide antibacterial properties and high osseointegration potential. This work reports the first attempt of Ga-species inclusion into an electrodeposited crystalline HAP surface. Such unique dual-function biointerfaces could show antimicrobial activity through the release of Ga ions, while tight bone bonding will be achieved through the apatite formed on the surface. The developed surface engineering strategy on Ti-45Nb alloy can be transferred to other biomedical alloy surfaces.

## 2. Materials and Methods

The substrate used in the present study is a cylindrical Ti-45Nb (wt.%, ASTM B348-13) alloy (diameter 1.5 cm) purchased from ATI specialty alloys and components (Dallas, TX, USA), which was cut into discs of thickness 0.1 cm. Microstructural characterization of the studied alloy was conducted by X-ray diffraction (XRD, STOE Stadi P diffractometer, STOE&Cie GmbH, Germany) and scanning electron microscopy (SEM, Zeiss Leo Gemini 1530, Zeiss Group, Germany). Comprehensive information about the alloy preparation and microstructural characterization techniques is provided elsewhere [26,28]. Electrochemical experiments were employed to investigate the corrosion behavior of β-type Ti-45Nb alloys. The alloy samples were mechanically ground with SiC emery paper (from grit 320 to grit 4000). Electrochemical measurements were performed in a phosphate-buffered saline (PBS) solution with a pH of 7.4. A Solartron XM ModuLab potentiostat (AMETEK Scientifuc Instruments, Oak Ridge, TN, USA) was used in combination with a three-electrode electro-chemical cell. Ti-45Nb alloys were used as the working electrode, and a platinum net was used as the counter electrode. The potentials were measured versus the saturated calomel electrode (SCE, E(SCE) = 0.241 V vs. standard hydrogen electrode (SHE) at 25 °C) as the reference electrode. For electrodeposition, the Ti-45Nb discs were mechanically polished using SiC sheets (P400-1200 grit), followed by a thorough ultrasonic cleaning in acetone, ethanol, and distilled water in succession for a period of 10 min each, and air-dried.

### 2.1. Electrodeposition of HAP and Ga(NO$_3$)$_3$ Immersion

The electrodeposition studies were conducted in an electrolyte comprising 0.00167 mol/L of $Ca(NO_3)_2 \cdot 4H_2O$ and 0.001 mol/L of $NH_4H_2PO_4$ diluted in deionized water. The pH of the solution was maintained at 5.4 at 296 K (adjusted by adding NaOH solution). A three-electrode electrochemical cell with a Ti-45Nb substrate as the working electrode, the Ag/AgCl electrode (SSE, E = 0.197 V vs. NHE) as a reference electrode, and a platinum mesh as a counter electrode was used, which was connected to a potentiostat (Bio-Logic S.A.S, Seyssinet-Pariset, France). The in-house built specimen holder for Ti-45Nb disc samples has an exposed area of 0.95 cm$^2$. The double-walled glass cell was connected to a thermostat to control the electrodeposition temperature of 333 K and the electrolyte was stirred with a magnetic stirrer. Potentiostatic deposition was done at a potential of −1.2 V vs. Ag/AgCl for a time period of 3 h, stepped up after open circuit potential (OCP) stabilization for 10 min (based on [15,40]). The current density transients were recorded. Afterward, the samples were cleaned by rinsing in distilled water and blow-dried.

To attain the goal of inclusion of $Ga^{3+}$ ions, wet chemical studies were conducted by immersing the HAP-coated Ti-45Nb samples in 1 mM of gallium nitrate, Ga(NO$_3$)$_3$ solution for 24 h. A low molarity was chosen as higher Ga ion contents can render an acidic nature to the solution which destabilizes and detaches the HAP coating from the titanium alloy surface. Hence, in order to maintain a pH value in the range of approximately 6.5–6.9, different buffers were added to 1 mM of Ga(NO$_3$)$_3$ which includes, phosphate, sodium bicarbonate, ammonium acetate, and citrate buffers (details enlisted in Table 1). In addition, the other criterion for selecting these buffers is based on their biocompatibility aspects,

which if present in trace amounts on the modified surface should not alter the favorable cellular interactions.

**Table 1.** Details of buffers used and sample codes.

| Sample Code | Buffer Used | Chemical Formula | Concentration | pH Range | Provider |
|---|---|---|---|---|---|
| $Ga_{phos}$ | phosphate | (I) $KH_2PO_4$ / (II) $Na_2HPO_4$ | 20 mM | 6.5–6.6 | Merck Millipore, Darmstadt, Germany |
| $Ga_{carb}$ | sodium bicarbonate | $NaHCO_3$ | 20 mM | 6.8–6.9 | Merck Millipore, Darmstadt, Germany |
| $Ga_{acet}$ | ammonium acetate | $CH_3COONH_4$ | 20 mM | 6.6–6.7 | Merck Millipore, Darmstadt, Germany |
| $Ga_{citr}$ | citrate | $HOC(CO_2H)(CH_2CO_2H)_2$ | 20 mM | 6.5–6.6 | VWR Chemicals, Leuven, Belgium |

*2.2. Surface Characterization*

The morphology of the developed surfaces post-electrodeposition and after the wet chemical step was analyzed by means of scanning electron microscopy (SEM) (GEMINI Leo 1530 Zeiss Group, Germany),. The phase analysis of the developed surfaces was conducted using a grazing incidence angle X-ray diffractometer (GI-XRD) with an incident angle of $1°$ utilizing Cu-K$\alpha$ radiation, 40 mA, 40 kV (Philips X'Pert Pro thin film diffractometer, Malvern Panalytical, Malvern, UK). The elemental distribution in the deposited surfaces along depth depth-wise direction was determined using a Glow Discharge Optical Emission Spectrometer (GD-OES) (GDA750HR Spectruma Analytik GmbH, Spectruma Analytik GmbH, Germany), equipped with a Universal Sample Unit capable of applying a radiofrequency voltage to produce sputtering craters in the range of 2.5 mm. Due to the roughness of the HAP-coated samples, the universal sample unit was used, where the sealing to the atmosphere is done outside the sample. Discharge parameters of anode voltage 500 V of tube generator and 2.7 hPa Ar pressure were used to analyze the samples to record spectral emission lines corresponding to Ti 399 nm, Ga 417 nm, Ca 393 nm, P 177 nm, Nb 316 nm, O 130 nm, H 121 nm, and Na 588 nm. Post-processing of the obtained GD-OES data was conducted to subtract the background signal of GaI 417.2062 nm developed due to elemental spectral lines of ArII 417.2067 nm and TiII 417.19038 nm in close proximity. An approximate coating thickness is calculated based on the sputtering rate of 67 nm/s. It must be mentioned that the obtained signal intensities are only directly correlated with the concentration of the analyzed elements if the sputtering rates and discharge conditions are constant.

The elemental concentration of the deposited HAP layer and the Ga incorporation after immersion in a 1 mM of Ga nitrate solution at pH 6.5 stabilized by different buffer agents were determined by inductively coupled plasma with optical emission spectrometry, ICP-OES (iCAP 6500 Duo View, Thermo Fisher Scientific, Waltham, MA, USA). The HAP layer was dissolved with 0.3 mL of half-concentrated $HNO_3$ (65%, analytical grade, Merck Millipore, in a 25 mL beaker. The digestion solution was filled up to 15 mL. The weight of the Ti-45Nb discs was determined before and after the digestion procedure with an analytical balance. The ICP-OES method was calibrated with five standard solutions based on commercial single-element standard solutions of 1 g/L (Carl Roth GmbH + Co. KG, Germany). The elemental concentrations of Ca, P, and Ga were measured at different wavelengths. Each analytical solution was measured three times. The average of the three single values was used for further calculations. Between two analytical solutions, a standard solution with known element concentrations was measured for drift correction.

**3. Results and Discussion**

Initial experiments to investigate the effect of metallurgical Ga-addition (up to 8 wt.%) on Ti-45Nb alloy on corrosion resistance, tribo-corrosion aspects, and antibacterial performance were recently conducted in our group [26–29]. It was proved that Ga addition

increases corrosion resistance by creating a very stable passive film and also offers a promising antibacterial effect. However, the naturally passivated surfaces of Ti alloys are bioinert.

The present work explores the possibilities of incorporating $Ga^{3+}$ ions into electrodeposited HAP coatings to have a unique combination of antimicrobial and bioactive properties of the Ti-45Nb alloy. This alloy composition was selected due to its reduced Young's modulus (E = 64 GPa) [28] compared to those of established Ti-implant materials (E = 110 GPa). A lower elastic modulus is associated with a more reduced stiffness, which is closer to that of human bone and is therefore beneficial for the reduction of stress shielding effects [41]. The reduced elastic modulus is due to the single-phase beta structure. The X-ray diffraction pattern of the Ti-45Nb is shown in Figure 1a, revealing a fully β-phase structure with no diffraction peaks associated with secondary phases (ICDD no-04-017-4957, bcc-Ti). The inset shows an SEM image illustrating the microstructure composed of equiaxed β-phase grains. In addition, this binary beta-type alloy exhibits excellent corrosion resistance, indicative of low corrosion activity and stable anodic passive film formation, as shown in Figure 1b.

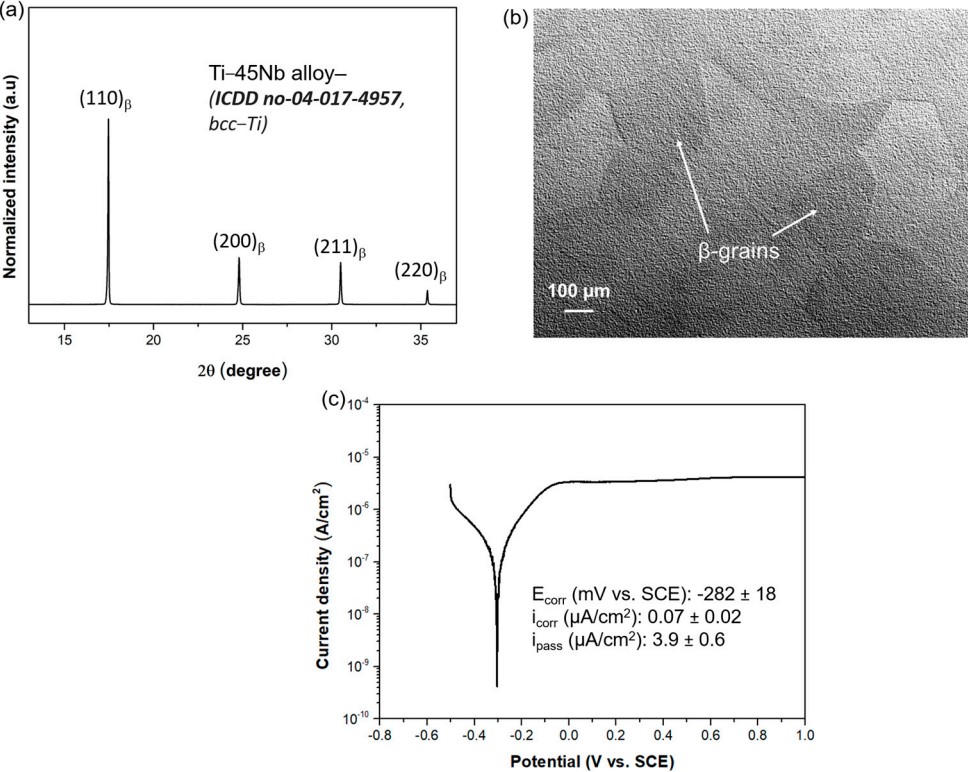

**Figure 1.** (**a**) X-ray diffraction pattern of Ti-45Nb alloy depicting a single body-centered cubic β-phase, (**b**) SEM image of the Ti−45Nb with equiaxed microstructure and (**c**) potentiodynamic polarization curve of Ti−45Nb alloy in PBS solution, with calculated corrosion potential ($E_{corr}$), corrosion and passivation current density ($i_{corr}$ and $i_{pass}$) values.

The excellent chemical stability of single phase β-Ti-45Nb brings new challenges to chemical surface modification treatments, which are indispensable for inducing osteogenesis and antibacterial performance. In this study, we used a novel approach to introduce additional biofunctionality and make the binary alloy bioactive by a Ga-doped coating, produced via a two-step process.

In this approach, an initial electrodeposition of HAP is followed by wet chemical reactions in the presence of $Ga(NO_3)_3$ to include Ga species on the surface. The surface chemistry and topography of a metallic substrate are expected to be decisive factors for an effective electrodeposition process [15]. Previous works have shown that chemical surface modifications, like oxidation or etching treatments, which are well established for cp-Ti and

Ti-6Al-4V, could not simply be transferred to beta-Ti-Nb alloys. Recent works [15,40,42] showed that the high Nb content has a significant effect on surface reactivity, mainly by enhancing surface passivity with the incorporation of Nb oxides. Based on the results obtained from the cyclic cathodic polarization studies [15,40], a suitable cathodic potential is determined for the electrodeposition of HAP in the aqueous solution of $Ca(NO_3)_2 \cdot 4H_2O$ and $NH_4H_2PO_4$. The potential of $-1.2$ V vs. Ag/AgCl is selected for the electrodeposition (comparatively more positive than the region that is dominated by the water reduction reaction). The details of cyclic cathodic polarization studies with stepwise reaction stages leading to HAP formation have been reported elsewhere [40,43]. For the potentiostatic deposition, the electrode potential was stepped up by raising from OCP to the deposition potential of $-1.2$ V vs. Ag/AgCl, and the corresponding current transient curve is displayed in Figure 2. An initial double-layer recharge followed by a gradual current density decay can be observed, similar to that reported by Schmidt et al. [40]. The observed current density maximum can be mainly ascribed to the superimposed steps of phosphate ion generation in addition to nitrate (and oxygen) reduction. The absolute value of current density decreases slowly, and it is attributed to the decrease in effective electrode surface area, which might be due to the chemical precipitation that blocks the Ti-45Nb surface with growing non-conductive calcium phosphate [40].

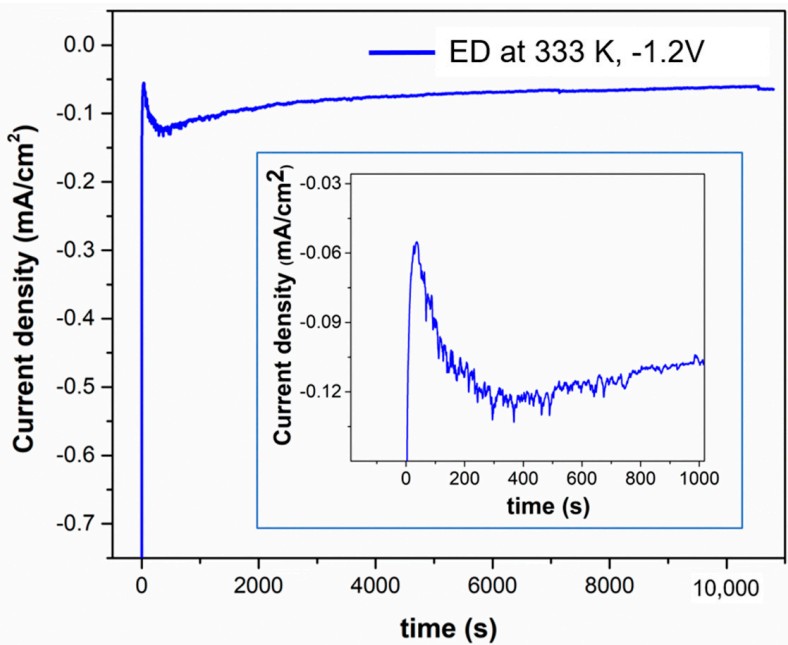

**Figure 2.** The current density transient of potentiostatic polarization conducted at $-1.2$ V at 333 K in electrolyte 0.00167 mol/L of $Ca(NO_3)_2 \cdot 4H_2O$ and 0.001 mol/L of $NH_4H_2PO_4$ solution, inset displays the initial transient period.

A limited layer growth is expected due to the retardation of further reduction reactions owing to the development of initial calcium-phosphate-based layers formed on the Ti-45Nb substrate surface. The observed layer after the electrodeposition is compact, dense, and uniformly distributed. As previously reported [40], brittle deposits with plate-like morphologies in sub-micrometer dimensions are observed during SEM analysis as displayed in Figure 3a. After a subsequent wet chemical step in $Ga(NO_3)_3$ solution for 24 h, the morphology of the coating is clearly retained in the case of using three buffers (Table 1)—phosphate (Figure 3b), sodium bicarbonate (Figure 3c) and ammonium acetate (Figure 3d). Only a slight thinning of the platelets is visible in the case of the edges of the sodium bicarbonate buffer. The corresponding EDS analysis is provided in Figure S1.

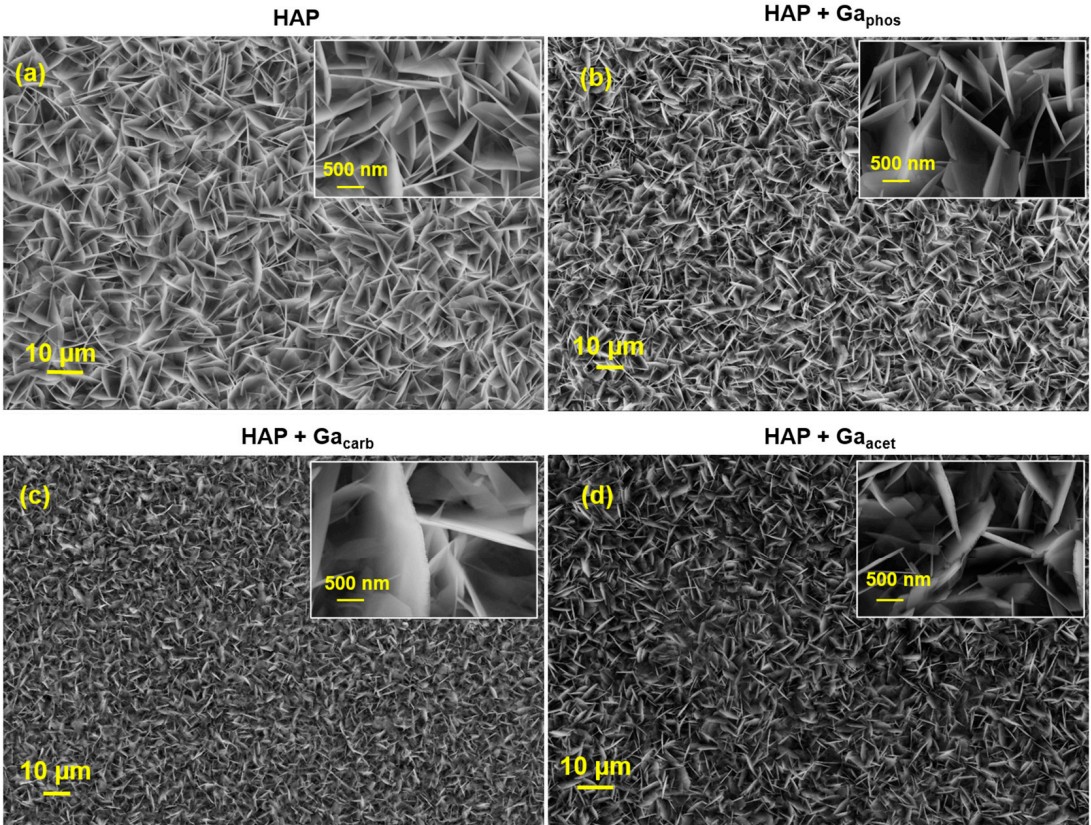

**Figure 3.** SEM images of (**a**) potentiostatically deposited HAP at −1.2 V vs. Ag/AgCl and Ga-HAP morphologies in the presence of (**b**) phosphate, (**c**) sodium bicarbonate, and (**d**) ammonium acetate buffers. The inset shows the magnified views of the HAP plates.

Apart from these buffers, citrate buffer was also investigated during the second wet chemical step to induce Ga into the electrodeposited HAP layer. Citrate was selected as it is biocompatible and widely used for clinical applications, and citrate-buffered $Ga(NO_3)_3$ received Food and Drug Administration (FDA) approval for treating malignancy-associated hypercalcemia [44]. However, the initially electrodeposited HAP layer is not retained after keeping in contact with the citrate-base buffer. The SEM morphology shown in Figure 4 evidences the detachment of HAP layers in the case of wet chemical studies in the presence of citrate buffer. The underlying mechanism of coating spalling could be the complexation of Ga-citrate ($C_6H_5GaO_7$) based compounds formed, which were observed as white precipitates at the bottom of the tubes after the wet chemical step. The exact chemistry behind this compound formation needs to be explored.

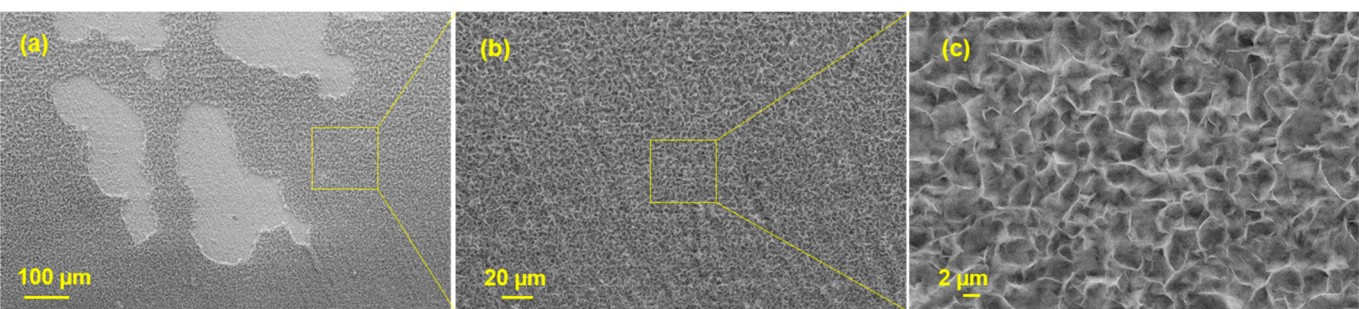

**Figure 4.** SEM images of (**a**–**c**) GA-HAP developed in the presence of citrate buffer showing the non-uniform coating with coating removal, and magnified views reveal the non-retainment of HAP morphology.

The phase composition of the developed surfaces was analyzed by means of the GI-XRD technique, and the respective patterns are displayed in Figure 5. The electrodeposited layer is composed of a well-crystallized single phase of hexagonal HAP (JCPDS No. 09-0432, space group P63/m) without the presence of any extraneous phase or peak shifts. Figure 5 displays the observed diffraction angles with the corresponding HAP Miller indices at 26.06° (002), 28.25° (102), 29.14° (210), 32.27° (300), 49.55° (213) and 53.5° (004). In addition to this, the typical body-centered-cubic (bcc) beta-Ti peaks at around 38.5° (110) and 55.56 (200) are observed arising from the substrate Ti-45Nb. The primary (002) reflection at approximately 26° indicates the preferred orientation of the developed plate-like HAP along (002) planes, pointing towards the growth of HAP plates growing with their c-axis perpendicular to the electrode surface. The presence of Ga species, as indicated by SEM, has not induced any remarkable phase changes in HAP.

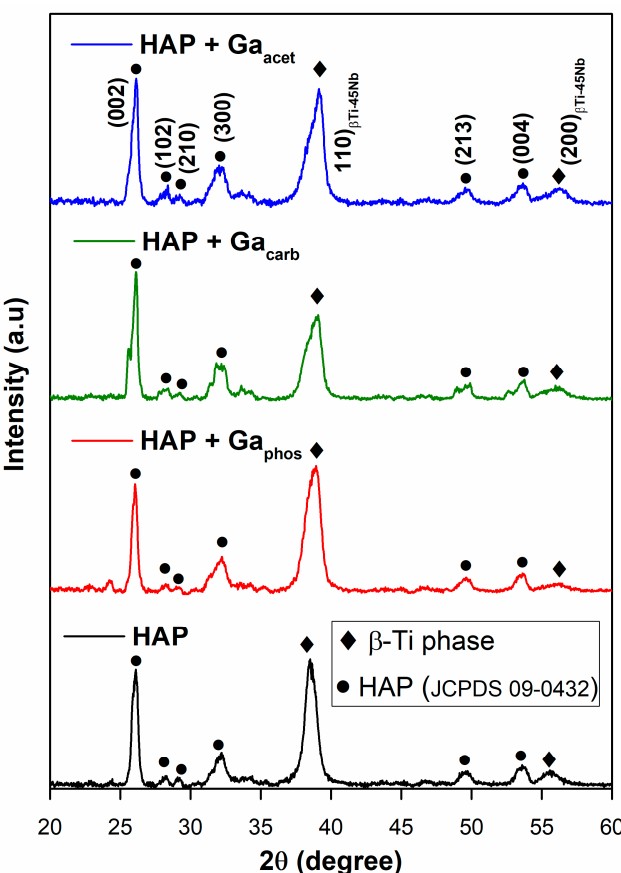

**Figure 5.** GI-XRD analysis of electrodeposited HAP surfaces, and Ga-HAP in the presence of phosphate, sodium bicarbonate, and ammonium acetate buffers.

In order to qualitatively evaluate the distribution of the main HAP constituents Ca, P, and O,H along the depth-wise direction of the developed surface states on Ti-45Nb substrates, and to confirm the presence of gallium species therein, a detailed GD-OES analysis was conducted, and the obtained depth profiles are shown in Figure 6. The presence of Ca, P, and Ga is observed on the electrodeposited surface which started to display a declining trend of surface element intensities (Ca, P, Ga, O, and H) along the HAP surface/Ti-45Nb interface followed by a simultaneous increase of elements of substrate material (Ti and Nb). Electrodeposited HAP surfaces have been detected with the presence of Ca and P on the surface (Figure 6a). As shown in Figure 6b–d, the presence of Ga on the developed HAP has been confirmed when using phosphate, sodium bicarbonate, and ammonium acetate buffers. Figure 6e confirms the relative trend of Ga distribution along the HAP surface. A denser HAP layer with Ga presence near the substrate surface can be

inferred owing to the increasing intensities of Ca, P, and Ga near the interface. The estimated coating thickness for the HAP layer is in the range of 2.68 ± 0.2 μm. It can also be noted from the depth profiles that a reduction in coating thickness occurs after the second step (due to the reduced time in the spectra to reach the substrate surface with dominant Ti and Nb spectra) after being immersed in 1 mM of Ga(NO$_3$)$_3$ for 24 h. The approximate coating thickness varied with different buffers in the order acetate (1.74 ± 0.23 μm) > carbonate (1.47 ± 0.18 μm) > phosphate (1.2 ± 0.11 μm).

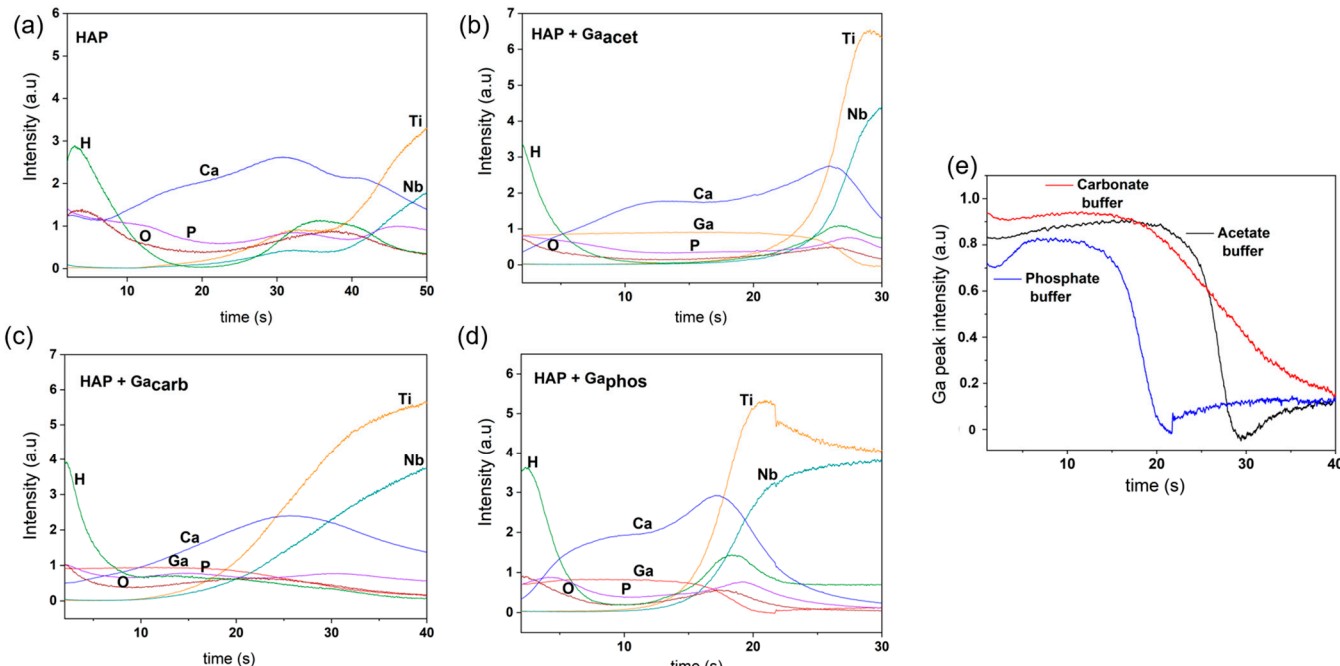

**Figure 6.** Glow discharge—optical emission spectrometry analysis of (**a**) potentiostatically deposited HAP surfaces, and Ga-modified surfaces in the presence of (**b**) phosphate, (**c**) sodium bicarbonate, (**d**) ammonium acetate buffers, and (**e**) Ga peak intensity observed with the three distinctive buffers (electrolyte temperature 333 K).

The bulk elemental composition of the developed coatings was assessed based on ICP-OES wet-chemical analysis. The electrodeposited HAP coating is composed of Ca and P with a Ca/P ratio of 1.4 ± 0.05 (Figure 7a) indicating the development of a Ca-deficient HAP which is not purely stoichiometric composition (for stoichiometric HAP, the Ca/P ratio is 1.67). One of the reasons could be the non-equilibrium conditions under which the electrodeposition of HAP takes place. This will hinder the formation of an energetically more favorable stoichiometric HAP. The other reason for the reduced Ca/P ratio could be due to the presence of a minor amount of Na ions. These ions introduced into the solution during the adjustment of the electrolyte pH are capable of substituting Ca ions in an HAP lattice [45]. Na is one of the principal constituents of natural bone; its presence poses no potential harm to the biocompatibility aspects of the developed surfaces. ICP analysis evidenced the presence of Ga in the electrodeposited layers after the second step, with about 1 wt.% Ga (on the HAP coating, relative to the molar concentration in the immersion solution) in all three coatings. Figure 7b gives an indication of the Ga incorporation in the sample prepared with acetate buffer, showing a comparatively improved Ga/(Ca + P) ratio (in percent), corroborating the trend observed during the GD-OES Ga distribution results for HAP + Ga$_{acet}$.

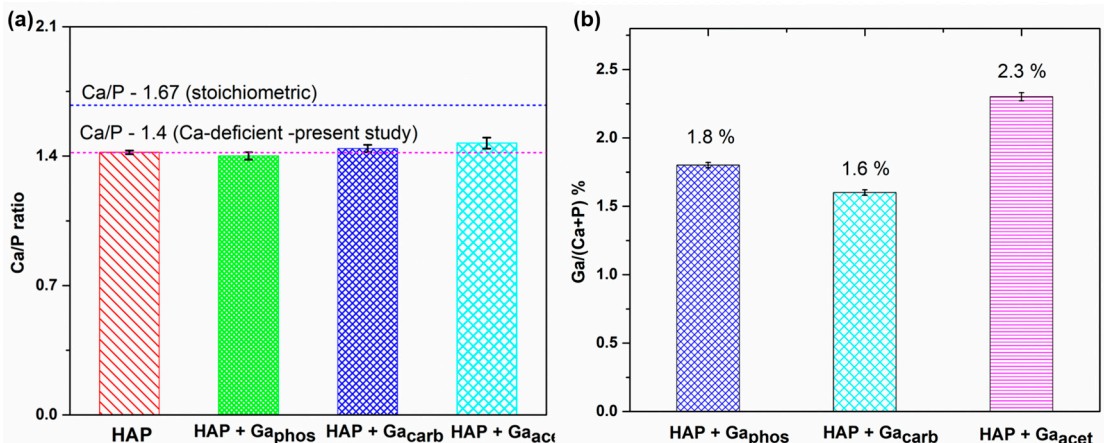

**Figure 7.** Composition analysis via the ICP-OES technique of the developed surfaces shows (**a**) the Ca/P ratio of the surfaces and (**b**) the percent of the Ga/(Ca + P) ratio.

It is observed that, by the second wet chemical step $Ga^{3+}$ is incorporated only in small quantities in the pre-formed HAP coating. This could be related to the distribution of $Ga^{3+}$ species on the surface of the large areas of the plate-like HAP crystals. Those HAP precipitates are very stable with low solubility in near-neutral aqueous media. Non-stoichiometric (Ca-deficient) hexagonal HAP crystals can accommodate monovalent or divalent cations in their crystal structure on $Ca^{2+}$ sites. The incorporation of $Ga^{3+}$ ions may be due to surface instabilities of the HAP crystal, thereby enabling dynamic ion exchange in near-surface regions [45]. Since SEM imaging (Figure 3) of those two-step-treated states did not reveal substantial changes in the HAP morphology, precipitation of Ga-compound layers onto HAP surfaces is not considered (or may be restricted to the nanoscale).

Several works have reported the relevance of pH value in controlling the stability of HAP coatings [32,46]. In the present study, 1 mM of $Ga(NO_3)_3$ was used for the wet chemical immersion studies. The pH value of the 1 mM of $Ga(NO_3)_3$ solution was maintained at around 6.5–6.9 with the addition of buffers such as phosphate, sodium bicarbonate, and ammonium acetate (20 mM), which yielded uniform coatings with 1 wt.% of $Ga^{3+}$ on the HAP surface. Kurtjak et al. [32] reported the achievement of optimal biological response for HAP/$Ga^{3+}$ powder when the $Ga^{3+}$ content is about 4 wt.%. The present work shows progress in terms of incorporating $Ga^{3+}$ (1 wt.%) species into an electrodeposited HAP coating, which needs to be improved for effective translative research. It is very critical to maintain a minimum amount of Ga on the modified surface, to ensure optimal antibacterial activity without impacting normal cells. Further research is needed to elucidate the Ga release kinetics, antibacterial, and biocompatibility aspects of the developed HAP-based electrodeposited layers.

## 4. Summary and Conclusions

The reported work explored the development of a novel two-step technique to incorporate $Ga^{3+}$ species into the HAP coating on low-modulus Ti-45Nb alloy. The plate-like morphology of electrodeposited HAP was retained after immersion studies in 1 mM gallium nitrate in the presence of phosphate, sodium bicarbonate, and ammonium acetate buffers (20 mM). $Ga^{3+}$ presence on the surface of large areas of the plate-like HAP crystals was confirmed by GD-OES analysis. In addition, GD-OES detected the HAP coating thickness in the range of $2.68 \pm 0.2$ μm and it was inferred that a reduction of coating thickness occurred after the immersion studies in gallium nitrate with different buffers in the following order: acetate ($1.74 \pm 0.23$ μm) > carbonate ($1.47 \pm 0.18$ μm) > phosphate ($1.2 \pm 0.11$ μm). A detailed ICP-OES analysis revealed a Ga content of 1 wt.% in the developed Ga-HAP surfaces. The Ca/P ratio was determined to be 1.4 in the electrodeposited HAP and after the Ga incorporation step.

In conclusion, the two-stage electrochemical-chemical approach is promising for incorporating Ga ion species in relevant quantities into HAP coatings. However, there are still many aspects to be explored, most importantly the identification of the optimal Ga ion content in implant surface coatings and adequate Ga ion release rates in the physiological environment in a post-operative scenario. Those should induce antibacterial activity and concurrently should not induce any cytotoxic effects. Further analysis of the release kinetics of the $Ga^{3+}$ ions will provide more insights into the antibacterial mechanism of these coatings. On the contrary, apart from the release kinetics, the contact killing mechanism is another proposed antibacterial mechanism for Ga, in which case the developed stable crystalline HAP with Ga will be beneficial.

**Supplementary Materials:** The following supporting information can be downloaded at: https://www.mdpi.com/article/10.3390/coatings13101817/s1, Figure S1. EDX analysis of (a) hydroxyapatite surface showing the presence of Ca, P on Ti-45Nb surface, and Ga-incorporated HAP morphologies in the presence of (b) phosphate, (c) sodium bicarbonate and (d) ammonium acetate buffers evidencing Ga addition on to the HAP-modified surface.

**Author Contributions:** Conceptualization, J.V., A.G. and M.C.; formal analysis, J.V., A.V., V.H. and L.A.A.; data curation, J.V., A.V., V.H., L.A.A. and A.A.; writing—original draft preparation, J.V.; writing—review and editing, A.G., L.A.A., A.A., B.S. and M.C.; visualization, J.V., L.A.A. and A.A.; supervision and funding acquisition, A.G. and M.C. All authors have read and agreed to the published version of the manuscript.

**Funding:** J.V. is grateful for the financial support from IFW Dresden awarding a fellowship for visiting scientists. L.A.A. and A.A. acknowledge funding from the European Commission within the H2020-MSCA grant agreement No. 861046 (BIOREMIA-ITN).

**Institutional Review Board Statement:** Not applicable.

**Informed Consent Statement:** Not applicable.

**Data Availability Statement:** The data presented in this study are available on request from the corresponding author.

**Acknowledgments:** The authors would like to acknowledge Kerstin Hennig and Anne Voidel for technical support.

**Conflicts of Interest:** The authors declare no conflict of interest.

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
