# Peer review of "Designing Gallium-Containing Hydroxyapatite Coatings on Low Modulus Beta Ti-45Nb Alloy"

_coatings, doi:10.3390/coatings13101817_

Round 1
Reviewer 1 Report
This work reported a method to prepare gallium-containing hydroxyapatite coatings on low modulus beta Ti-45Nb alloy. The materials were well charactered and the results are good. I recommend it for publication after moderate editing of English language.
Moderate editing of English language required.
Author Response
Response: We would like to thank the reviewer for recommending the manuscript for publication. As suggested by the reviewer, English language has been thoroughly checked throughout the manuscript and necessary edits are amended.
Reviewer 2 Report
The authors should add in all the XRD spectra the JCPDS No. 273 09-0432. In Figure 1, the SEM image must be changed (it is unclear).
The EDS spectra together with the elemental mapping results should be added in the manuscript. The EDS quantitative results should be added in the manuscript.
Please add the results of the coatings adhesion test together with the SEM measurements of the coatings thickness.
What about surface wettability of the coatings?
There are to much self-citations (for example : 8 references from 46).
The results of the FTIR measurements should be added in the manuscript.
In my opinion, after the authors bring all these improvements to the manuscript, it can be considered for publication in Coatings.
Author Response
Q1. The authors should add in all the XRD spectra the JCPDS No. 273 09-0432.
Response: As recommended by the reviewer, the JCPDS card number of HAP has been incorporated in the XRD pattern as shown in Figure 5 (Page 8, Line 303). The updated Figure 5 is in the attached pdf.
Q2. In Figure 1, the SEM image must be changed (it is unclear).
Response: The SEM image in Figure 1 a inset shows the microstructure of Ti-45Nb alloy, which corresponds to an equiaxed single phase beta body-centred cubic (b-bcc) structure. As pointed out by the reviewer, the SEM has been modified for an improved visibility as shown in the attached pdf (Page 5, Line 218).
Figure 1. a) X-ray diffraction pattern of Ti-45Nb alloy depicting a single body-centered cubic b-phase, b) SEM image of the Ti-45Nb with equiaxed microstructure and c) potentiodynamic polarization curve of Ti-45Nb alloy in PBS solution, with calculated corrosion potential (Ecorr) , corrosion and passivation current density (icorr and ipass) values
Q3. The EDS spectra together with the elemental mapping results should be added in the manuscript. The EDS quantitative results should be added in the manuscript.
Response: As recommended by the reviewer, the EDS spectra of the developed coatings has been provided as a supplementary file (Figure S1). We would like to convey here that EDS is a semi-quantitative elemental analysis technique and is not recommended for quantitative elemental analysis in a comparatively rough surface like the HAP surface we have obtained. Since the sample surface is not homogenous with respect to the beam interaction volume, the elemental analysis results and the elemental mapping will vary based on the contribution from the neighbouring components.
Q4. Please add the results of the coatings adhesion test together with the SEM measurements of the coatings thickness.
Response: The adhesion strength of electrodeposited HAP has been reported by our research group in previous work [1]. A scratch test of HAP layer/substrate system revealed a critical loading of 9 ± 0.76 N for electrodeposited HAP. It is a very good recommendation by the reviewer to elucidate how gallium addition will affect this value; we will explore this aspect in a future study. In the present work, the thickness of the developed coatings was estimated by using Glow Discharge - Optical Emission Spectroscopy (GD-OES) technique. Problems were encountered during Focused Ion Beam (FIB)-SEM analysis due to the presence of gallium in the developed coating.
[1] Schmidt, R.; Hoffmann, V.; Helth, A.; Gostin, P.F.; Calin, M.; Eckert, J.; Gebert, A. Electrochemical deposition of hydroxyapatite on beta-Ti-40Nb. Surface and Coatings Technology 2016, 294, 186-193, doi:https://doi.org/10.1016/j.surfcoat.2016.03.063.
Q5. What about surface wettability of the coatings?
Response: Surface wettability is one of the very important properties to be considered while developing an implant coating. However, due to the inherent porosity of the developed HAP coating, the water droplets will be absorbed into the HAP layer without leaving a water envelope for wettability measurement. Complete wetting induced by the HAP surface is also attributed to the abundance of hydroxyl (OH-) polar groups on the surface.
Q6. There are too much self-citations (for example: 8 references from 46).
Response: The reference list consists of 7 references from the works by our research group. We would like to justify here that these cited works are very relevant and novel works in the research direction of Ga-containing beta Ti alloys and electrodeposition-based surface modification strategy to incorporate Ga into HAP coatings. Hence in order to provide an overall idea of the works in this specific research field, these references are cited in appropriate places.
Reference numbers 15, 40, 42- related to electrodeposition of HAP
Reference numbers – 26, 27,28,29- related to Ga incorporation into beta Ti alloys.
Q7. The results of the FTIR measurements should be added in the manuscript.
Response: We would like to convey here that the main motive of the present work is to investigate the inclusion of Ga ions into the HAP coating. Hence, we have focused in the present work to identify the Ga presence in the electrodeposited HAP coatings via various techniques such as Glow Discharge - Optical Emission Spectroscopy (GD-OES) and Inductively Coupled Plasma Optical Emission Spectrometry (ICP-OES). We would like to thank the reviewer for valuable suggestion regarding the use of FTIR characterization. Fourier transform infrared (FTIR) is a widely used technique for the identification of functional groups in organic compounds where a permanent dipole moment exists; this can be used to identify the phosphate and hydroxyl groups in the case of hydroxyapatite. In order to gain more insights into this aspect using FTIR and to understand structural modification due to Ga incorporation in HAP using detailed Nuclear magnetic resonance spectroscopy (NMR) spectroscopy, we will add these in-depth characterizations as a separate paper in our future work.
In my opinion, after the authors bring all these improvements to the manuscript, it can be considered for publication in Coatings.

Reviewer 3 Report
The manuscript "Designing gallium-containing hydroxyapatite coatings on low modulus beta Ti-45Nb alloy" by J. Vishnu et al. covers a very useful and applicable topic on implant materials. There is a constant need to improve existing and develop new implant materials to reduce problems caused by the formation of bacterial biofilms or the degradation of materials from which implants are made.
In the focus is a modification of the alloy based on the coating of hydroxyapatite and galium ions. Gallium and its compounds have recently attracted much interest due to their beneficial pharmacological properties and safety for humans. The study is a logical continuation of investigations focusing on galium and its properties for implant applications that have recently been published in several papers.
The manuscript is well and clearly written, the experimental part is detailed enough and the results are clearly presented.
The recommendation is a minor revision along the lines of the following:
-page 3, lines 122-123: Please add the potential value of the reference electrode used.
-page 5, lines 200-201, Fig.1a): Please add the JCPDS card used to identify the crystal structure.
-Fig.6b): Please correct. Ga ions should be shown.
Author Response
Response: We would like to the thank for the positive comments and recommendation for publication of the manuscript.
The recommendation is a minor revision along the lines of the following:
Q1. Page 3, lines 122-123: Please add the potential value of the reference electrode used.
Response: We would like to thank the reviewer for pointing out this missing aspect. The potential value of the reference electrode has been updated in the revised manuscript (Page 3, Line 125) as provided below.
‘The potentials were measured versus the saturated calomel electrode (SCE, E(SCE) = 0.241 V vs. standard hydrogen electrode (SHE) at 25°C) as the reference electrode.’
Q2. Page 5, lines 200-201, Fig.1a): Please add the JCPDS card used to identify the crystal structure.
Response: As suggested by the reviewer, the corresponding XRD database card for body centred cubic beta Ti data from the International Centre of Diffraction Data (ICDD) PDF2 database (ICDD no-04-017-4957, bcc-Ti) has been added in the revised manuscript (Page 5, Line 213) as follows.
‘The X-ray diffraction pattern of the Ti-45Nb is shown in Figure 1a, revealing a fully β-phase structure with no diffraction peaks associated with secondary phases (ICDD no-04-017-4957, bcc-Ti)‘.
Q3. Fig.6b): Please correct. Ga ions should be shown.
Response: We are extremely thankful to the reviewer for pointing out this mistake. The Ga ions have been correctly marked in the updated manuscript (Page 9, Line 327) as shown in the attached pdf.

Round 2
Reviewer 2 Report
accept in present form